# TRIM24 is an oncogenic transcriptional co-activator of STAT3 in glioblastoma

Deguan Lv[1,2], Yanxin Li[3], Weiwei Zhang[1], Angel A. Alvarez[4], Lina Song[1], Jianming Tang[1], Wei-Qiang Gao[1], Bo Hu[4], Shi-Yuan Cheng[1,4] & Haizhong Feng [1]

Aberrant amplification and mutations of epidermal growth factor receptor (EGFR) are the most common oncogenic events in glioblastoma (GBM), but the mechanisms by which they promote aggressive pathogenesis are not well understood. Here, we determine that non-canonical histone signature acetylated H3 lysine 23 (H3K23ac)-binding protein tripartite motif-containing 24 (TRIM24) is upregulated in clinical GBM specimens and required for EGFR-driven tumorigenesis. In multiple glioma cell lines and patient-derived glioma stem cells (GSCs), EGFR signaling promotes H3K23 acetylation and association with TRIM24. Consequently, TRIM24 functions as a transcriptional co-activator and recruits STAT3, leading to stabilized STAT3-chromatin interactions and subsequent activation of STAT3 downstream signaling, thereby enhancing EGFR-driven tumorigenesis. Our findings uncover a pathway in which TRIM24 functions as a signal relay for oncogenic EGFR signaling and suggest TRIM24 as a potential therapeutic target for GBM that are associated with EGFR activation.

[1] State Key Laboratory of Oncogenes and Related Genes, Renji-Med X Clinical Stem Cell Research Center, Ren Ji Hospital, School of Medicine, Shanghai Jiao Tong University, Shanghai 200127, China. [2] School of Biomedical Engineering, Shanghai Jiao Tong University, Shanghai 310000, China. [3] Key Laboratory of Pediatric Hematology and Oncology Ministry of Health, Pediatric Translational Medicine Institute, Shanghai Children's Medical Center, School of Medicine, Shanghai Jiao Tong University, Shanghai 200127, China. [4] Department of Neurology, Northwestern Brain Tumor Institute, The Robert H. Lurie Comprehensive Cancer Center, Northwestern University Feinberg School of Medicine, Chicago, IL 60611, USA. Deguan Lv and Yanxin Li contributed equally to this work. Correspondence and requests for materials should be addressed to H.F. (email: Fenghaizhong@sjtu.edu.cn)

Glioblastoma (GBM) is the most common malignant primary brain cancer of adults with a grim median survival of 14.6 months upon diagnosis[1,2]. Epidermal growth factor receptor (EGFR) amplification and mutations are major drivers promoting glioma tumor growth and invasion through persistent activation of signaling networks and metabolic reprogramming[3]. Recent global genomic and transcriptome analyses reveal EGFR-induced signaling with epigenetic remodeling[4]. However, the mechanisms by which EGFR controls the transcriptional machinery through epigenetic modification are not well known.

Post-translational modifications (PTMs) of histone proteins play pivotal roles in many cellular processes, including transcription[5,6]. Histones can be covalently modified by a variety of chemical alterations, including methylation and acetylation[6]. Because acetylation can neutralize the positive charge of lysine residues, it was initially proposed that acetylated proteins promote an open chromatin structure by weakening the association of the negatively charged DNA with the protein core of the nucleosome[7]. Subsequent work identified acetylated proteins that are bound by acetyl lysine reader proteins containing binding bromodomain (BRD), demonstrating that PTM can also exert its effect by recruiting chromatin binding proteins to regulate various cellular functions[5,6]. Although a large body of knowledge had been accumulated about the characteristics and biological functions of histone acetylation, the mechanisms by which they contribute to cancer are largely unknown.

TRIpartite Motif-containing protein 24 (TRIM24), also known as Transcription Intermediary Factor 1 alpha (TIF1α) is a reader of non-canonical histone signature H3K23ac[8]. TRIM24 has amino-terminal RBCC domains (Ring, BBox and Coiled-Coil), characteristic of the TRIM family of proteins, and a TIF1 subfamily-defining plant homeodomain (PHD)-bromodomain[9]. TRIM24 has been shown to function as an oncogene or tumor suppressor dependent on the context. Although genomic deletion of mouse TRIM24 promotes hepatocellular carcinoma (HCC)[10,11], aberrant overexpression of human TRIM24 is positively correlated with cancer progression and poor survival of patients in multiple cancers, including gastric cancer[12], bladder cancer[13], non-small cell lung cancer[14], human HCC[15], head and neck carcinoma[16] and breast cancer[8,17]. TRIM24 also functions as an E3 ligase to target p53 in Drosophila and human breast cancer[18]. TRIM24 was identified as a transcription cofactor of receptors such as estrogen receptor (ER) in breast cancer[8] and androgen receptor (AR) in prostate cancer[19] to interact with chromatin and these nuclear receptors via its tandem

PHD-bromodomain binding to H3K23ac, leading to activation of downstream signaling related with tumor progression. However, the function of TRIM24 in cancers is still largely unknown. Here, using RNA-Seq and chromatin immunoprecipitation-quantitative real-time PCR (ChIP-qRT-PCR) analyses of GBM cell lines, patient-derived glioma stem cells (GSCs) and clinical GBM specimens, we identify a novel signaling pathway whereby EGFR-upregulated H3K23ac binds with TRIM24, and TRIM24 functions as a co-activator to recruit STAT3, leading to stabilized STAT3-chromatin interactions and subsequent activation of STAT3 downstream signaling, thereby enhancing EGFR-driven tumorigenesis.

## Results

**EGFR specifically upregulates H3K23ac expression in gliomas.** To determine roles of histone modification in EGFR-driven gliomagenesis, we analyzed expression of histone H3 lysine 23 acetylation (H3K23ac), histone H3 lysine 27 trimethylation (H3K27me3), histone H3 lysine 4 trimethylation (H3K4me3) and histone H3 lysine 27 acetylation (H3K27ac)-four histone modifications associated with transcriptional regulation[8,19–23] using Western blotting in isogenic U87 and LN229 GBM cells with, or without, stable expression of the ligand-independent activated EGFR mutant, EGFRvIII. This analysis revealed that H3K23ac was significantly upregulated in EGFRvIII-expressing GBM cells compared with the controls, whereas expression of H3K27me3, H3K4me3, and H3K27ac were not affected (Fig. 1a). In U87 GBM cells with stable overexpression of EGFR, EGF stimulation also markedly increased H3K23ac expression with no effects on expression levels of H3K27me3, H3K4me3 and H3K27ac compared to the controls, respectively (Fig. 1b). The treatment with the EGFR tyrosine kinase inhibitor, erlotinib significantly inhibited H3K23ac expression stimulated by EGF, whereas there were no effects on the levels of H3K27me3, H3K4me3 and H3K27ac (Fig. 1b). This data demonstrates that activated EGFR specifically upregulates H3K23ac in GBM cells.

To test whether EGFRvIII's effect on H3K23ac expression depended on its kinase activity, we used a kinase dead EGFRvIII construct (EGFRvIII-KD; Fig. 1c)[24–27]. In U87 GBM cells, EGFRvIII, but not EGFRvIII-KD, significantly elevated H3K23ac expression levels without altering the expression of H3K27me3, H3K4me3 and H3K27ac (Fig. 1c), indicating that the observed effect of EGFRvIII on H3K23ac expression was not due to exogenous EGFRvIII expression, and the effect of EGFRvIII on H3K23ac expression was dependent on EGFRvIII kinase activity.

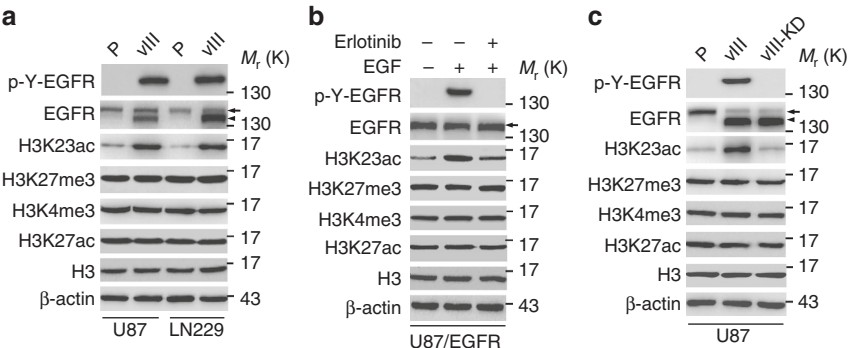

**Fig. 1** EGFR signaling enhances H3K23ac expression in GBM cells. **a** Effects of EGFRvIII overexpression on histone H3 methylation and acetylation in U87 and LN229 GBM cells. P, parental cells; vIII, U87 or LN229 cells expressing EGFRvIII. **b** Erlotinib treatment inhibited EGF-stimulated EGFR phosphorylation and H3K23 acetylation in U87 GBM cells stably expressing EGFR. EGF (20 ng/ml) stimulated U87 GBM cells with, or without, erlotinib (10 μM) for 24 h. **c** Western blotting (WB) analysis of the effects of kinase dead EGFRvIII (EGFRvIII-KD) on H3K23ac expression. Data in **a–c** are representative of three independent experiments. β-actin and histone H3 (H3) were used as loading controls. Arrows, EGFR. Arrow heads, EGFRvIII

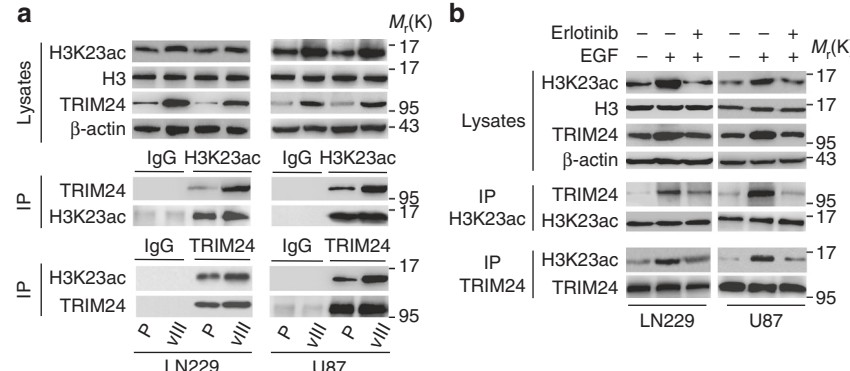

**Fig. 2** EGFR promotes the association between H3K23ac and TRIM24. **a** Immunoprecipitation (IP) and WB analyses of the association between H3K23ac and TRIM24 in LN229 and U87 GBM cells with, or without, EGFRvIII overexpression. **b** EGFR signaling increases the association of H3K23ac and TRIM24, which is attenuated by the inhibitor erlotinib. EGF (20 ng/ml) stimulated U87 GBM cells with, or without, erlotinib (10 μm) for 24 h. Data in **a** and **b** are representative of three independent experiments. β-actin, IgG and H3 were used as controls

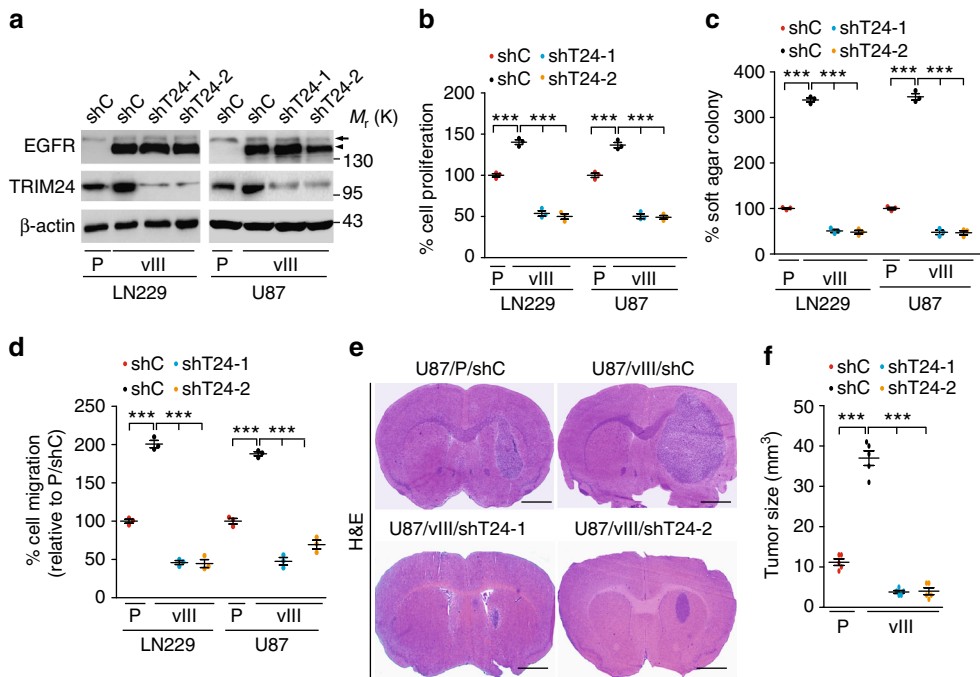

**Fig. 3** Knockdown of TRIM24 inhibits EGFR-driven cell proliferation, cell survival, colony formation in soft agar, cell migration, and tumor growth. **a** WB assays of *TRIM24* knockdown with two different shRNAs (shT24-1 and shT24-2) or a control shRNA in LN229 and U87 GBM cells with, or without EGFRvIII overexpression. Arrows, EGFR. Arrow heads, EGFRvIII. **b–d** Effects of TRIM24 knockdown by shT24 or shC on cell proliferation (**b**), colony formation in soft agar (**c**) and cell migration (**d**) in vitro. **e** Knockdown of TRIM24 inhibits brain xenograft growth of EGFRvIII-promoted U87 glioma cells. Representative images of H&E stained brain tumor xenograft with indicated U87 GBM cells. Scale bars: 1 mm. **f** Quantification of tumor size. Data were from H&E stained brain sections of five mice per group of two independent experiments. Error bars, s.d. Data represent two or three independent experiments with similar results. ***$P < 0.001$, paired two-way Student's *t*-test, compared with parental or EGFRvIII cells or tumors treated with shC

**EGFR promotes the association of H3K23ac and TRIM24.**
TRIM24 is a reader of non-canonical histone signature H3K23ac[8], and has been reported to function as an oncogene in various human cancers[8,12–17], including gliomas[28]. To gain insight into the role of TRIM24 in gliomagenesis, we first performed expression analysis using the Oncomine database[29] and found that expression levels of *TRIM24* mRNA were upregulated in GBM compared with normal brain tissues in Murat Brain dataset[30] (Supplementary Fig. 1a) and The Cancer Genome Altas (TCGA) data set (Supplementary Fig. 1b). Moreover, expression levels of *TRIM24* mRNA were higher in GBM than high grade, low grade and normal brain tissues in the GSE4290 data set[31] (Supplementary Fig. 1c). These data support that TRIM24 is important for gliomagenesis.

Next, to investigate the impact of TRIM24 on EGFR-driven glioma tumorigenesis, we assessed the expression of *TRIM24* mRNA in isogenic U87 and LN229 GBM cells with, or without, stable expression of EGFRvIII. As shown in Supplementary Fig. 2, *TRIM24* mRNA expression levels were significantly increased in GBM cells transduced with EGFRvIII compared to the controls. EGFRvIII-upregulated TRIM24 protein expression also was confirmed by western blotting assays (Fig. 2a).

Given the previously reported function of TRIM24 to regulate cancer progress through interacting with H3K23ac[8,19], we further detected the interaction between TRIM24 and H3K23ac, and revealed that they are bound in U87 and LN229 GBM cells (Fig. 2a). Moreover, EGFRvIII markedly promoted TRIM24

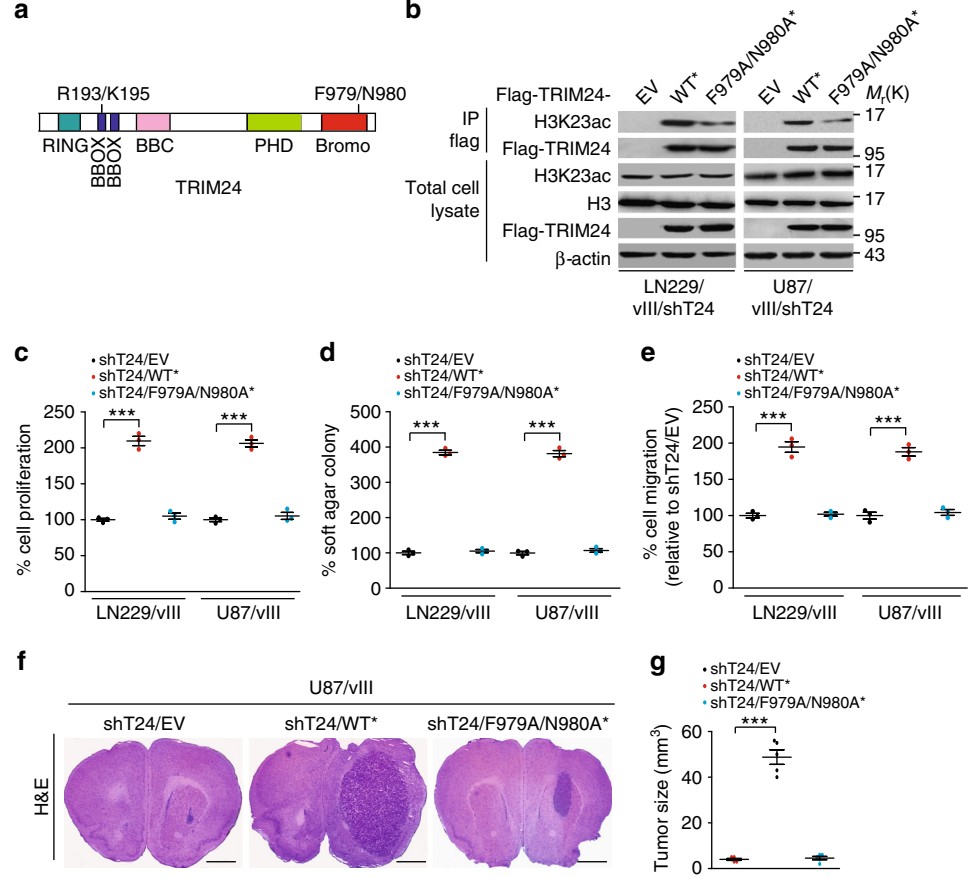

**Fig. 4** Binding of TRIM24 with H3K23ac is required for EGFR-driven tumor growth. **a** Schematic of TRIM24. RING, RING domain; BBOX, B box domain; PHD, plant homeodomain; Bromo, Bromodomain. **b** Re-expression of Flag-TRIM24 shRNA-resistant TRIM24$^{WT*}$, but not TRIM24$^{F979A/N980A*}$ mutant (F979A/N980A*) or an empty vector control (EV), restores the association between H3K23ac and TRIM24 in LN229/vIII/shT24 and U87/vIII/shT24 GBM cells. **c–e** Re-expression of Flag-TRIM24 shRNA-resistant TRIM24$^{WT*}$, but not TRIM24$^{F979A/N980A*}$ mutant rescues EGFRvIII- stimulated cell proliferation (**c**), colony formation in soft agar (**d**) and cell migration (**e**) in TRIM24 knockdown EGFRvIII glioma cells. **f** Re-expression of Flag-TRIM24 shRNA-resistant TRIM24$^{WT*}$, but not TRIM24$^{F979A/N980A*}$ mutant rescues EGFRvIII-stimulated tumor growth in TRIM24 knockdown EGFRvIII glioma brain xenografts. Images represent results of 5 mice per group of 2 independent experiments. Scale bars: 1 mm. **g** Quantification of tumor size. Data were from stained brain sections of five mice per group of two independent experiments. Error bars, s.d. Data represent two or three independent experiments with similar results. ***$P < 0.001$, paired two-way Student's $t$-test, compared with EGFRvIII cells or tumors treated with shT24/EV

association with H3K23ac in both U87 and LN229 GBM cells (Fig. 2a). EGF stimulation in U87 and LN229 GBM cells increased TRIM24 expression and its association with H3K23ac, and the treatment with the EGFR tyrosine kinase inhibitor erlotinib attenuated both (Fig. 2b). These data suggest that TRIM24 is upregulated by activated EGFR, and that EGFR signaling enhances the association between TRIM24 and H3K23ac in GBM cells.

**Depletion of TRIM24 inhibits EGFR-driven tumor growth.**
Although it was reported to be important for gliomagenesis[28], the role of TRIM24 in glioma has not been fully elucidated. We and others have demonstrated that the constitutively active, ligand-independent EGFRvIII promotes U87 and LN229 tumorigenesis in orthotopic xenograft models[25,26,32]. To test whether TRIM24 is critical for EGFRvIII-driven glioma tumorigenesis, we knocked down TRIM24 in LN229/EGFRvIII and U87/EGFRvIII cells using two separate shRNAs (Fig. 3a). Compared to the controls, depletion of TRIM24 in EGFRvIII-expressing cells markedly inhibited EGFRvIII-promoted cell proliferation (Fig. 3b), colony formation in soft agar (Fig. 3c) and cell migration (Fig. 3d) in vitro. When various engineered U87 cells were implanted into the brains of animals, knockdown of endogenous TRIM24 by two separate

shRNAs significantly reduced EGFRvIII-stimulated tumor growth relative to non-silencing control xenografts (Fig. 3e, f). Importantly, shRNA knockdown of endogenous TRIM24 in LN229 and U87 GBM cells had no effect on the expression of EGFR, EGFRvIII, or β-actin, thus excluding off-target effects of the shRNA constructs and confirming our hypothesis that TRIM24 is a downstream target of EGFR signaling (Fig. 3a). Collectively, these findings suggest that TRIM24 plays an important role in EGFR/EGFRvIII-driven tumorigenesis in gliomas.

**Binding of TRIM24 with H3K23ac is required for tumor growth.** Binding of TRIM24 with H3K23ac is required for cell proliferation in breast cancer and prostate cancer[8,19], and phenylalanine 979 (F979)/asparagine 980 (N980) of TRIM24 in the bromodomain is critical for their association[8] (Fig. 4a). To assess whether of the association of TRIM24 with H3K23ac is required for glioma tumorigenesis, we conducted complementation experiments in LN229 and U87 GBM cells with TRIM24 shRNAs. We engineered GBM cells expressing shRNA-resistant TRIM24 encoding the wild type (WT*) and F979A/N980A mutant (F979A/N980A*). Re-expression of TRIM24-WT* rescued the association between TRIM24 and H3K23ac (Fig. 4b), cell proliferation (Fig. 4c), colony formation in soft agar (Fig. 4d)

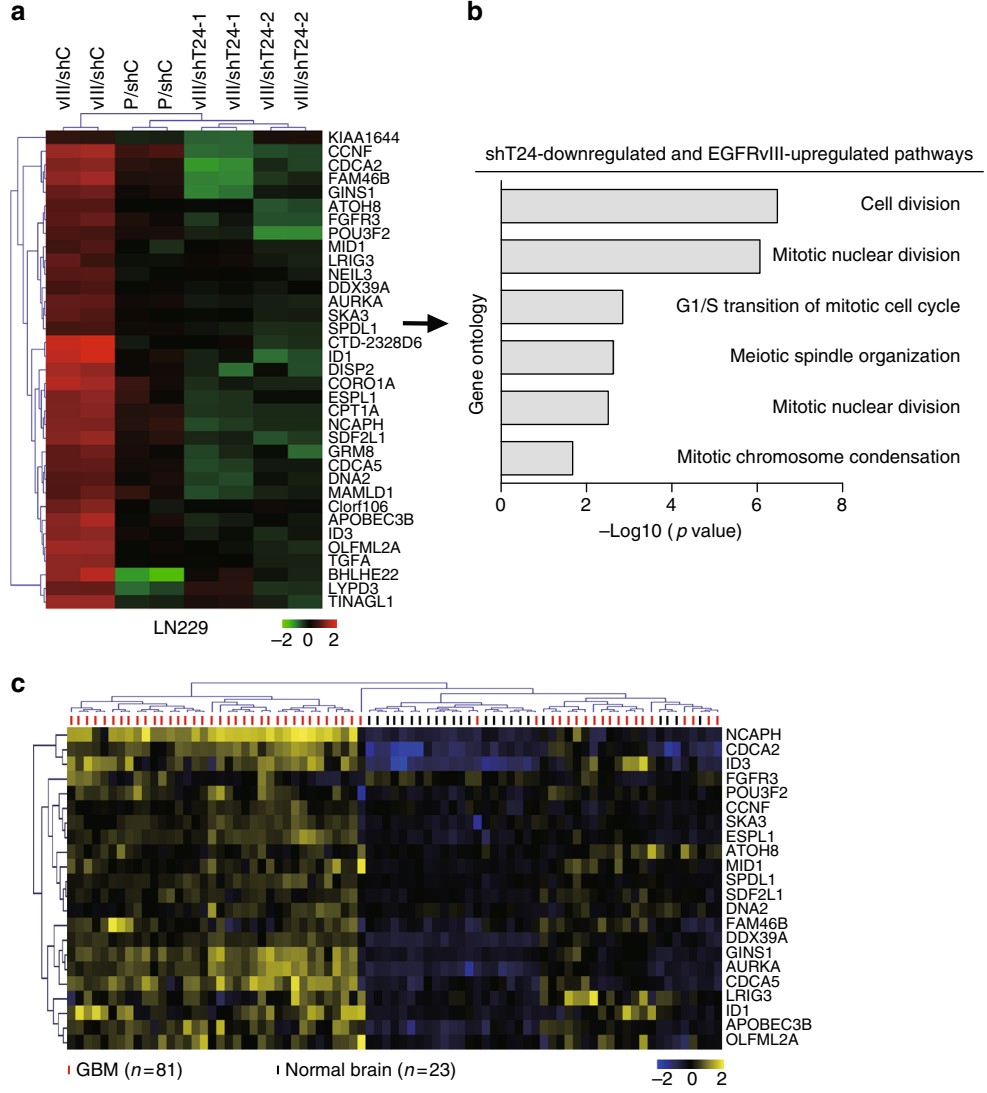

**Fig. 5** TRIM24 and EGFR co-upregulated genes are highly expressed in clinical GBM specimens. **a** Heatmap of mRNA-Seq analysis of differentially expressed genes (2-fold change and FDR <0.05) among LN229/P/shC, LN229/vIII/shC, LN229vIII/shT24-1 and LN229vIII/shT24-2 GBM cells. TRIM24 and EGFR-regulated genes were defined to be upregulated by EGFRvIII overexpression and downregulated by knockdown of TRIM24. **b** Gene ontology (GO) analysis indicates that genes upregulated by EGFRvIII and downregulated by TRIM24 knockdown are associated with cell proliferation pathways. **c** Hierarchical clustering analysis of patient gene expression data (GSE4290) indicates that the 22-gene signature resulted in two main clusters indicated in blue and yellow, respectively. Expression data of 23 non-tumors (brain tissues from epilepsy patient) and 81 GBM were downloaded from GSE4290[31]

and cell migration (Fig. 4e) *in vitro*. However, re-expression of TRIM24-F979A/N980A* did not fully restore the association of TRIM24 and H3K23ac (Fig. 4b), cell proliferation (Fig. 4c), colony formation in soft agar (Fig. 4d) and cell migration (Fig. 4e). We further injected engineered U87/EGFRvIII/TRIM24 shRNA cells transduced with an empty vector (EV), TRIM24-WT* or -F979A/N980A* into the mouse brain tumor xenograft. As shown in Fig. 4f, g, re-expression of TRIM24-WT* rescued EGFRvIII-driven tumor growth, whereas re-expression of TRIM24-F979A/N980A* could not. These data demonstrated that the association of TRIM24 with H3K23ac is important for EGFRvIII-driven tumorigenesis in gliomas.

**TRIM24 and EGFR-upregulated pathways are highly expressed.** To understand why TRIM24 is important for EGFR/EGFRvIII-driven glioma tumorigenesis, we performed transcriptome analysis in LN229 parental (LN229/P) and LN229/EGFRvIII (LN229/vIII) GBM cells transfected with an empty

vector or TRIM24 shRNAs (LN229/vIII/shT24-1 and LN229/vIII/shT24-2) using RNA-seq. This analysis identified 340 genes whose expression was markedly reduced by *TRIM24* knockdown with two different TRIM24 shRNAs in LN229/EGFRvIII GBM cells (fold change >2, p < 0.05; Supplementary Fig. 3a). These 340 genes were highly overrepresented in gene ontologies that are associated with cell proliferation pathways (Supplementary Fig. 3b). This data is consistent with our results and other reports that TRIM24 is important for cancer cell proliferation and tumor growth, including gliomas[28].

To further assess which genes are co-upregulated by EGFR and TRIM24 in glioma cells, we identified 35 genes that were significantly reduced by *TRIM24* knockdown and upregulated by EGFRvIII in LN229 GBM cells (fold change >2, p < 0.05; Fig. 5a). These 35 genes were enriched for pathways associated with cell proliferation pathways (Fig. 5b). RNA-Seq results for EGFRvIII-upregulated and TRIM24 shRNA-downregulated genes were validated by measuring mRNA levels on a subset of six selected genes involved in tumor growth by quantitative

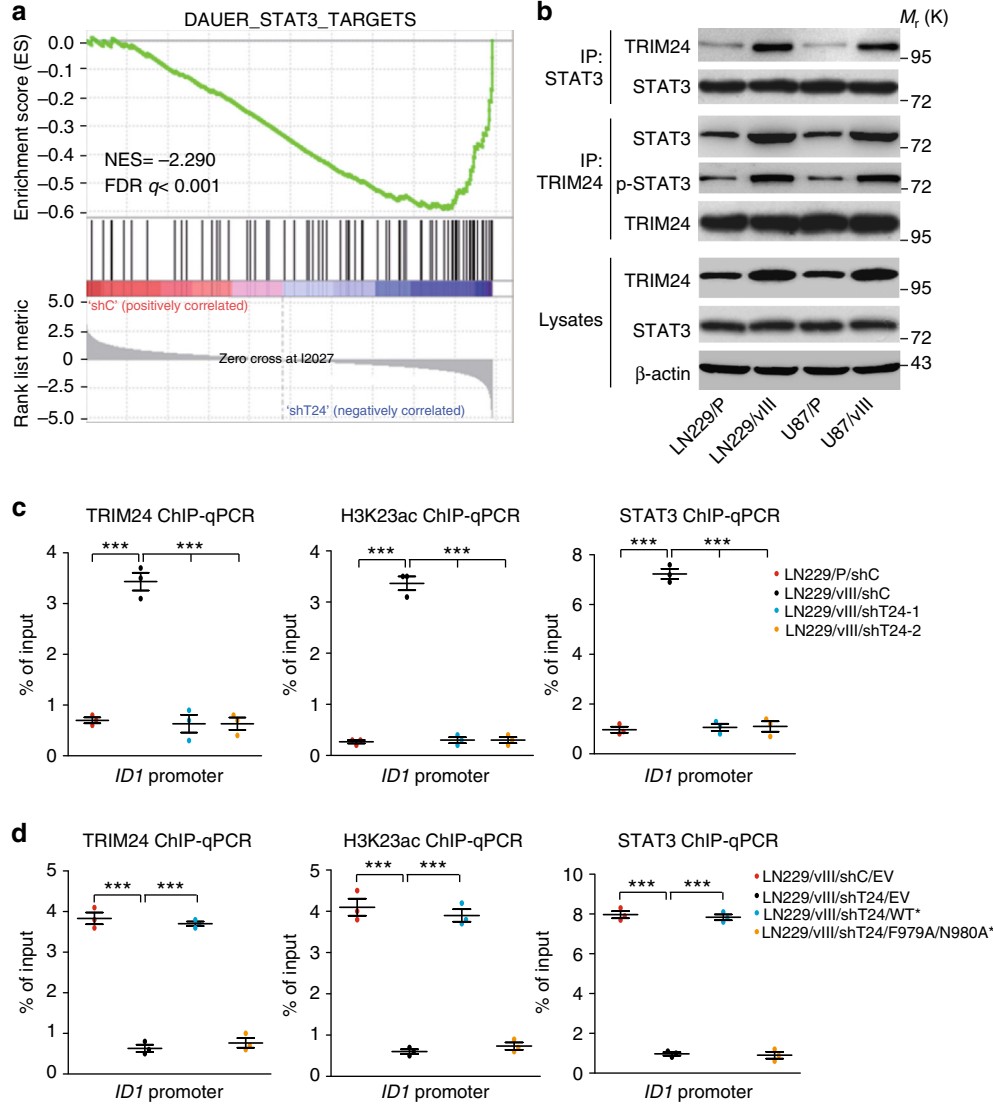

**Fig. 6** TRIM24 functions as a co-activator and stabilizes STAT3-chromatin interactions. **a** Gene set enrichment analysis of STAT3 target genes[34] using ranked gene expression changes in *TRIM24* knockdown LN229/EGFRvIII cells compare to control cells. NES, normalized enrichment score. **b** IP and WB analyses of TRIM24 and STAT3 expression and binding in LN229 and U87 GBM cells with, or without EGFRvIII overexpression. **c** ChIP-qRT-PCR analyses of the binding of TRIM24, STAT3 and H3K23ac with the *ID1* promoter. **d** ChIP-qRT-PCR analyses of the effects of Flag-TRIM24 shRNA-resistant TRIM24[WT*], or TRIM24 [F979A/N980A*] mutant or an empty vector control (EV), on the binding of TRIM24, STAT3 and H3K23ac with ID1 promoter. Error bars, s.d. Data represent two or three independent experiments with similar results. ***$P < 0.001$, paired two-way Student's *t*-test, compared with parental cells treated with shC or shC/EV

RT-PCR (qRT-PCR) analyses, including Inhibitor of DNA binding 1 (*ID1*), Inhibitor of DNA binding 3 (*ID3*), *Fibroblast growth factor receptor 3* (*FGFR3*), *Transforming growth factor alpha* (*TGFA*), Aurora kinase A (*AURKA*) and DexD-box helicase 39 A (*DDX39A*) (Supplementary Fig. 4). We then used this signature to stratify publicly available gene expression data from glioma patients[31], and found that 22 of our signature 35 genes were highly co-expressed in GBM specimens compared with normal brain tissues (Fig. 5c). Clustering analysis of these genes effectively segregated GBM from normal brain tissues, indicating the predictive value of our gene signature.

**TRIM24 functions as a transcriptional co-activator of STAT3.** Given the established role of TRIM24 protein as a transcriptional co-activator in breast cancer and prostate cancer[8,19], we hypothesized that TRIM24 functions as a transcriptional co-activator

in gliomas. To test our hypothesis, we performed gene set enrichment analysis (GSEA) in *TRIM24* knockdown LN229/ EGFRvIII GBM cells vs. control cells. As shown in Fig. 6a and Supplementary Fig. 5a, EGFR-regulated transcription factor *STAT3* and *NFKB* gene signatures[33,34] were significantly altered in *TRIM24* knockdown GBM cells. Then, we performed immunoprecipitation assays and found that TRIM24 binds with STAT3 and phospho-STAT3 (p-STAT3) (Fig. 6b) but not with the NF-κB subunit p65 (Supplementary Fig. 5b) in LN229 and U87 GBM cells. Moreover, TRIM24-STAT3 association was increased by EGFRvIII in two different GBM cells (Fig. 6b). STAT3 is a downstream effector of EGFR, activated in 60% of GBM patients, promotes progression in animal models of glioma, and negatively correlates with survival[32,35–38]. Taken together, our data suggest that TRIM24 is a co-activator of STAT3 in gliomas.

We further determined wether TRIM24 is transcriptionally regulated by STAT3. As shown in Supplementary Fig. 6a and b,

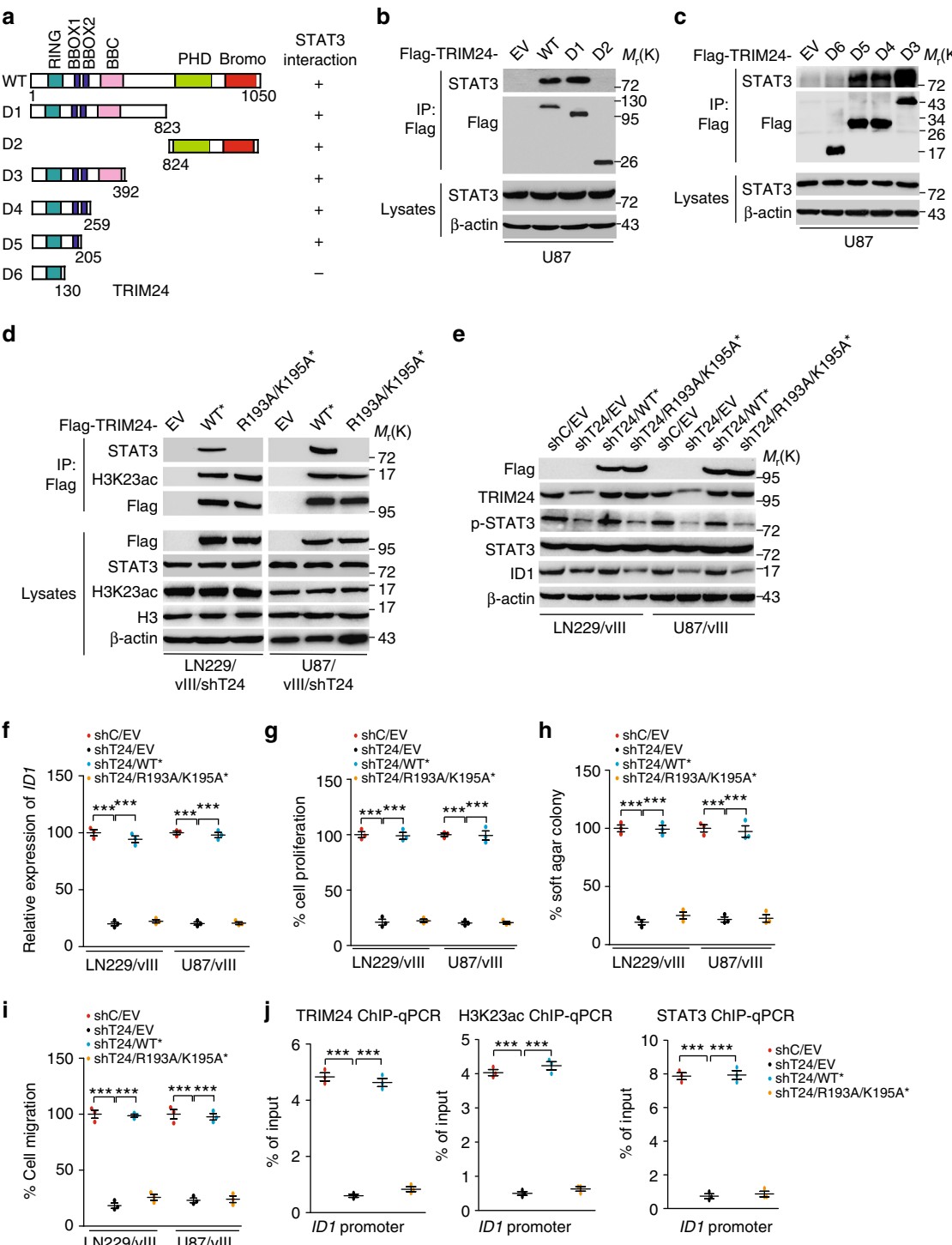

**Fig. 7** R193/K195 sites are necessary for TRIM24 to bind with and recruit STAT3 to chromatin. **a** Schematics of TRIM24$^{WT}$ and various TRIM24 deletion constructs. **b**, **c** STAT3 interacts with TRIM24 through its BBOX1 domain with amino acid residues 131–205. TRIM24$^{WT}$ or the indicated TRIM24 mutants were transfected into U87 GBM cells. **d** Re-expression of shRNA-resistant Flag-tagged TRIM24$^{WT*}$, but not TRIM24$^{R193A/K195A*}$ mutant or an empty vector control (EV), resulted in binding between TRIM24 and STAT3 in LN229/vIII/shT24 and U87/vIII/shT24 GBM cells. **e** Effects of re-expression of shRNA-resistant Flag-tagged TRIM24$^{WT*}$, TRIM24$^{R193A/K195A*}$ mutant, or an empty vector control (EV), on STAT3 phosphorylation and ID1 protein expression. **f** qRT-PCR analysis of effects of re-expression of shRNA-resistant TRIM24$^{WT*}$, TRIM24$^{R193A/K195A*}$ mutant or an empty vector control (EV) on *ID1* mRNA expression. **g**–**i** Effects of re-expression of shRNA-resistant TRIM24$^{WT*}$, TRIM24$^{R193A/K195A*}$ mutant or an empty vector control (EV) on cell proliferation **g**, colony formation in soft agar **h**, and cell migration **i**. **j** ChIP-qRT-PCR analyses of effects of re-expression of shRNA-resistant TRIM24$^{WT*}$, TRIM24$^{R193A/K195A*}$ mutant or an empty vector control (EV) on the binding of TRIM24, STAT3 and H3K23ac with the ID1 promoter. Error bars, s.d. Data represent two or three independent experiments with similar results. ***$P < 0.001$, paired two-way Student's *t*-test, compared with EGFRvIII cells treated with shC/EV

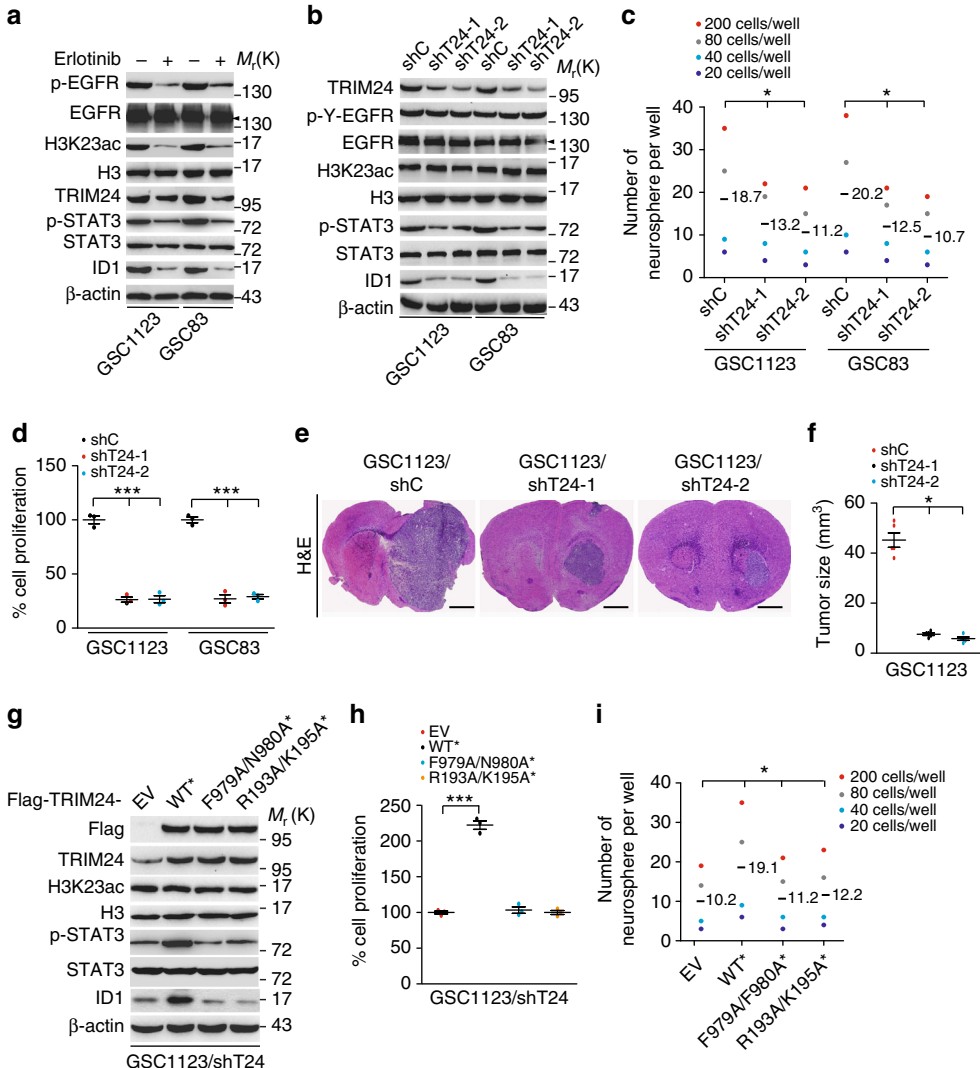

**Fig. 8** H3K23ac/TRIM24/STAT3-ID1 axis mediates EGFRvIII-driven glioma stem cell proliferation and self-renewal. **a** Effects of EGFRvIII on expression of H3K23ac, TRIM24, ID1, STAT3 and p-STAT3. GSC1123 and GSC83 GSCs were treated with Erlotinib (10 μM) for 24 h. β-Actin, EGFR or p-EGFR was used as a control. Arrow heads, EGFRvIII. **b** Knockdown of TRIM24 inhibits ID1 expression and STAT3 phosphorylation, but not H3K23ac expression, in GSC1123 and GSC83 GSCs. **c** Limiting dilution neurosphere-forming assays measuring the effects of TRIM24 knockdown. **d** Cell proliferation assays of effects of TRIM24 knockdown. **e** Representative images of H&E, analyses of brain sections, with indicated GSC1123 GSCs. Images represent results of 5 mice per group of 2 independent experiments. Scale bars: 1 mm. **f** Quantification of tumor size. Data were from stained brain sections of 5 mice per group of 2 independent experiments. **g, h** Effects of re-expression of Flag-TRIM24 shRNA-resistant TRIM24^WT*, TRIM24 ^F979A/N980A* mutant, TRIM24^R193A/K195A* mutant or an empty vector control (EV) on expression of H3K23ac, STAT3, ID1 and p-STAT3 (**g**) and cell proliferation (**h**) in GSC1123/shT24 GSCs. **i** Limiting dilution neurosphere-forming assays of effects of re-expression of Flag-TRIM24 shRNA-resistant TRIM24^WT*, TRIM24^F979A/N980A* mutant, TRIM24^R193A/K195A* mutant or an empty vector control (EV) on GSC1123/shT24 GSCs. Error bars, s.d. Data represent two or three independent experiments with similar results. *P < 0.05, ***P < 0.001, paired two-way Student's t-test, compared with GSC cells or tumors treated with shC or shT24/EV

STAT3 inhibitor cryptotanshinone (CTN)[39] treatment inhibited STAT3 phosphorylation, TRIM24 protein and *TRIM24* mRNA expression, but not EGFR/EGFRvIII expression and phosphorylation. *STAT3* knockdown with two different shRNAs also inhibited TRIM24 protein (Supplementary Fig. 6c) and *TRIM24* mRNA expression (Supplementary Fig. 6d), but no effects on EGFR/EGFRvIII expression and phosphorylation (Supplementary Fig. 6d). Moreover, overexpression of STAT3C (constitutively active STAT3) activated *TRIM24* promoter activity in U87 cells measured by promoter reporter assays (Supplementary Fig. 6e). This data shows that TRIM24 is transcriptionally mediated by STAT3 in EGFR/EGFRvIII-driven gliomas.

In the 22-gene signature, ID1, a target of STAT3[40,41], was upregulated by EGFRvIII and downregulated by TRIM24 knockdown in LN229 GBM cells (Fig. 5a, Supplementary

Fig. 7a and b). Overexpression of Flag-tagged ID1 rescued TRIM24 shRNA-inhibited cell proliferation and colony formation in soft agar in LN229/EGFRvIII GBM cells (Supplementary Fig. 7c–e). To further assess the role of TRIM24 as a transcriptional co-activator of STAT3, we performed ChIP-quantitative-PCR (ChIP-qPCR) on ID1 using antibodies directed against TRIM24, STAT3 and H3K23ac to measure binding to the promoter of *ID1* in LN229 GBM cells with or without EGFRvIII. As shown in Fig. 6c, TRIM24, STAT3 and H3K23ac all could bind to the *ID1* promoter. Compared with the control (P/shC), EGFRvIII significantly increased the binding of TRIM24, STAT3 and H3K23ac with *ID1* promoter. Knockdown of TRIM24 markedly impaired EGFRvIII-stimulated TRIM24, H3K23ac and STAT3 binding with *ID1* promoter (Fig. 6c). Moreover, re-expression of shRNA-resistant TRIM24 wild type (WT*)

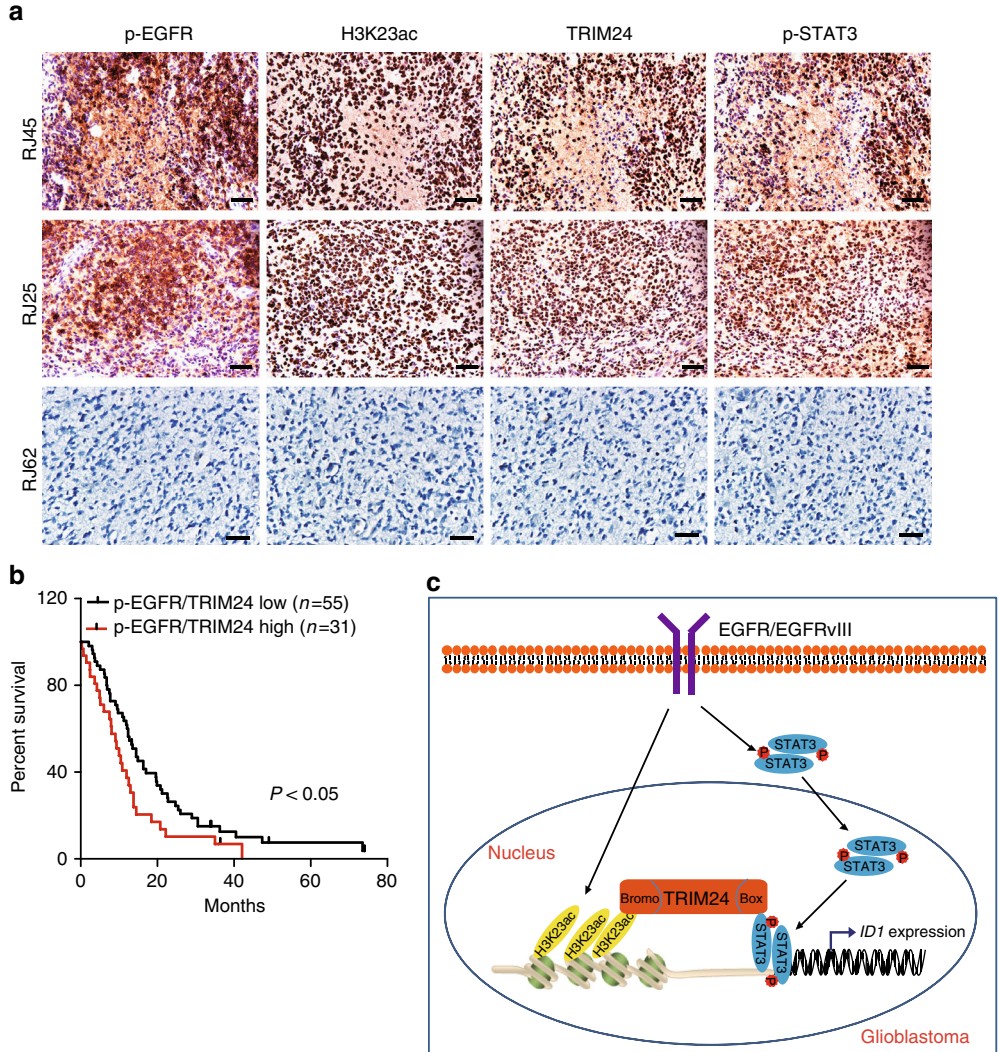

**Fig. 9** Co-expression of p-EGFR, H3K23ac, TRIM24, and p-STAT3 correlates with worse prognosis of GBM. **a** A total of 125 clinical GBM specimens were analyzed by IHC. Representative images of serial sections of a GBM tissue using anti-p-EGFR, anti-H3K23ac, anti-TRIM24, and anti-p-STAT3 antibodies are shown. Data are representative of two independent experiments. Scale bars: 25 μm. **b** Kaplan–Meier analyses of patients with high p-EGFR/TRIM24-expressing tumors (red line) vs. low p-EGFR/TRIM24-expressing tumors (black line) in IHC analyses. Median survival (in months): low, 14.5; high, 10.0. $P$ values were calculated by log-rank test. Black bars indicate censored data. **c** A working model of EGFR/H3K23ac/TRIM24/STAT3 signaling pathway in glioma tumorigenesis. EGFR-upregulated H3K23ac binds TRIM24 which recruits STAT3, leading to activation of STAT3 signaling, enhancing EGFR-driven GBM tumorigenesis

rescued the binding of TRIM24, STAT3 and H3K23ac with *ID1* promoter, whereas re-expression of shRNA-resistant TRIM24 mutant F979A/N980A (F979A/N980A*) did not (Fig. 6d). Taken together, our data demonstrate that TRIM24 recruits STAT3 to chromatin in EGFR-driven glioma cells, which is dependent on H3K23ac association, thus providing evidence that TRIM24 binds with EGFR-upregulated H3K23ac, and then recruits STAT3 and stabilizes its interaction with chromatin.

**R193/K195 sites are necessary for TRIM24-recruiting STAT3.**
To further identify which region or domain in TRIM24 mediates its association with STAT3, we generated several deletion mutants lacking various functional binding domains (Fig. 7a). We constructed TRIM24 C-terminus deletion mutant D1 and N-terminus deletion mutant D2, and separately transfected them into U87 GBM cells. Mutant D1, but not mutant D2 that contains only the C-terminal tandem plant Homeodomain (PHD) and bromodomain regions was able to bind with STAT3 (Fig. 7b),

suggesting that the N-terminus of TRIM24 is required for interacting with STAT3. Then, four N-terminal truncated mutants, D3 to D6, were constructed. When these N-terminal truncated mutants were separately transfected into U87 GBM cells, all except mutant D6 that contained only the RING domain were able to interact with STAT3 (Fig. 7c). This data suggests that the N-terminal region of amino acid residues 131–205 of TRIM24 protein is important for its association with STAT3.

To assess which sites of N-terminal region of amino acid residues 131–205 of TRIM24 are necessary for the association of TRIM24 and STAT3, we further constructed TRIM24 vectors with specific point mutations, arginine 193 (R193)/lysine 195 (K195) to alanine 193 (A193)/alanine (A195). Re-expression of shRNA-resistant TRIM24 cDNA encoding the WT* rescued the binding of TRIM24 with STAT3 and H3K23ac (Fig. 7d) in LN229/EGFRvIII and U87/EGFRvIII GBM cells. However, re-expression of a TRIM24 shRNA-resistant vector containing two-point mutations at R193 and K195 (R193A/K195A*) rescued the association of TRIM24 and H3K23ac, but did not restore the

binding of TRIM24 with STAT3 (Fig. 7d), suggesting that R193/K195 sites are critical for STAT3 binding.

Next, we detected effects of TRIM24$^{R193A/K195A*}$ mutant on STAT3 phosphorylation (p-STAT3), ID1 expression, cell proliferation, colony formation in soft agar and cell migration in LN229/vIII and U87/vIII GBM cells. Re-expression of TRIM24$^{WT*}$ rescued p-STAT3 level (Fig. 7e), the protein and mRNA expression levels of ID1 (Fig. 7e, f), cell proliferation (Fig. 7g), colony formation in soft agar (Fig. 7h) and cell migration (Fig. 7i) upregulated by EGFRvIII in GBM cells, whereas re-expression of TRIM24-R193A/K195A* could not (Fig. 7e–i). We also performed ChIP-qPCR on ID1 using antibodies directed against TRIM24, STAT3, and H3K23ac. As shown in Fig. 7j, re-expression of TRIM24$^{WT*}$ rescued binding of TRIM24, STAT3 and H3K23ac to ID1 promoter, whereas re-expression of R193A/K195A* mutation of TRIM24 did not. Together, our data demonstrate that TRIM24 association with STAT3 is important for STAT3 transcription activity and R193/K195 sites of TRIM24 are required for TRIM24 binding with STAT3.

**H3K23ac-TRIM24-STAT3-ID1 axis mediates GSC proliferation**. To further determine whether TRIM24 is critical for glioma tumorigenesis, we analyzed the effects of TRIM24 on patient-derived glioma stem cells (GSCs) using established methods to evaluate cell signaling pathways, self-renewal, proliferation, and tumor forming ability[26,42–45]. As shown in Fig. 8a, TRIM24 was highly expressed in both GSC83 and GSC1123 cells. These GSCs have been previously characterized by gene expression profiling as a mesenchymal subtype[42], express high levels of endogenous EGFRvIII[26], and are highly tumorigenic in orthotopic mouse xenografts[26,42]. Treatment of EGFR inhibitor, erlotinib, significantly inhibited EGFR phosphorylation (p-EGFR), STAT3 phosphorylation (p-STAT3), H3K23 acetylation, and expression of TRIM24 and ID1 (Fig. 8a). Knockdown of endogenous TRIM24 using two separate shRNAs in both GSC lines markedly suppressed p-STAT3 and ID1 expression (Fig. 8b), neurosphere formation (Fig. 8c), cell proliferation (Fig. 8d), and tumorigenesis of intracranial xenografts (Fig. 8e, f), validating our observations in U87/EGFRvIII (Fig. 3b–f).

To further validate the signal pathway of EGFR/H3K23ac/TRIM24/STAT3, we re-expressed different shRNA-resistant TRIM24 constructs, wild type (TRIM24$^{WT*}$), TRIM24 with H3K23ac binding site mutations F979A/N980A* (TRIM24$^{F979A/N980A*}$), and TRIM24 with STAT3 binding site mutations R193A/K195A* (TRIM24$^{R193A/K195A*}$) in GSC1123 cells with TRIM24 shRNAs. As shown in Fig. 8g–i, re-expression of shRNA-resistant TRIM24 wild type rescued p-STAT3, ID1 expression, cell proliferation and neurosphere formation, whereas neither re-expression of the mutation F979A/N980A nor the mutation R193A/K195A was able to restore the other effects. Collectively, these findings suggest that H3K23ac/TRIM24/STAT3 signal pathway plays an important role in EGFR/EGFRvIII-driven tumorigenesis in human gliomas.

**Co-expression of p-EGFR and TRIM24 is clinically prognostic**. Increased expression of EGFR, EGFRvIII, and p-STAT3 is closely associated with a poor prognosis for patients with malignant gliomas[32,35–38]. To further define the clinical relevance of our findings in this study, we examined expression of phospho-EGFR$^{Y1173}$ (p-EGFR)[26], H3K23ac, TRIM24 and phospho-STAT3$^{Y705}$ (p-STAT3) in clinical cancer samples. Using antibodies with validated specificities against these four proteins, we performed IHC analyses on serial sections of 125 clinical GBM specimens (Fig. 9a). In GBM tissues, co-expression of H3K23ac,

TRIM24, and p-STAT3 was found in the majority of p-EGFR positive tumors (Fig. 9a). Spearman's rank correlation analysis, based on quantification of the IHC staining (Supplementary Table 1)[27], showed that these correlations were statistically significant. Moreover, Kaplan-Meier analyses of survival showed that high expression levels of p-EGFR but not TRIM24 could serve as a predictor of a worse prognosis for patients with gliomas (Supplementary Fig. 8). Moreover, co-expression of p-EGFR and TRIM24 (Fig. 9b), co-expression of p-EGFR/H3K23ac, co-expression of TRIM24/H3K23ac or co-expression of TRIM24/p-STAT3 at high levels correlated with significantly shorter survival in patients with GBM (Supplementary Fig. 8). Taken together, these data support the role of EGFR/H3K23ac/TRIM24/STAT3 signaling in the pathophysiology, clinical progression, and aggressiveness of human gliomas. These results also suggest that TRIM24 in conjunction with other clinical markers could improve the assessment of clinical outcomes in GBM with EGFR activation.

## Discussion

Here, we reveal a novel mechanism of H3K23ac/TRIM24/STAT3-mediated gliomagenesis driven by EGFR activation. H3K23ac is non-canonical histone post-translational modification[8], and is associated with a poor prognosis in breast cancer[46] and oocyte polarization in *Drosophila*[47]. We show that H3K23ac is significantly and specifically upregulated by EGFR activation in GBM cells compared to other histone modifications tested. EGFR-upregulated H3K23ac binds with TRIM24, and then enhances TRIM24 recruitment of STAT3 to chromatin. To the best of our knowledge, this is the first demonstrating TRIM24 as an oncogenic transcriptional co-activator of STAT3, which enhances STAT3 transcription and promotes GBM tumorigenesis and GSC self-renewal, through a complex with H3K23ac (Fig. 9c). We identify the specific domains of TRIM24, particularly the bromodomain and N-terminal BBOX region, are critical in EGFR-driven tumorigenic activity. The clinical importance of our observations is strongly supported by the data showing that H3K23ac is co-expressed with p-EGFR, TRIM24, and p-STAT3 in clinical GBM specimens and co-expression of these markers correlates with shorter survival outcomes in GBM patients. Thus, this study provides clinical and mechanistic evidence demonstrating that H3K23ac/TRIM24 is critical for EGFR-driven tumorigenesis in human gliomas.

We identify TRIM24 as an oncogene in gliomas, consistent with previous reports[13,16]. Furthermore, we demonstrate that TRIM24 functions as a signal relay in mediating EGFR/EGFRvIII-driven tumorigenesis. We show that TRIM24 is upregulated and amplified in GBM specimens, and its expression is correlated with glioma progression. TRIM24 is upregulated by EGFR activation, and is required for EGFR-driven glioma cell proliferation, cell migration, colony formation in soft agar, orthotopic xenograft tumor growth in mouse brain. Our analyses of clinical gliomas reveal a close correlation between co-expression of TRIM24/p-EGFR, TRIM24/H3K23ac and TRIM24/p-STAT3 with poor prognoses in GBM patients. In addition, we also identify a 22-gene signature co-upregulated by EGFR and TRIM24 in GBM specimens using RNA-Seq analysis, which further support that TRIM24 facilitates EGFR-driven gliomagenesis.

Our study suggests that TRIM24 association with H3K23ac is required for EGFR/EGFRvIII-driven tumorigenesis. TRIM24 was demonstrated to bind with H3K23ac via its tandem PHD-bromodomain in breast cancer[8] and prostate cancer[19]. Here, in agreement with previous reports, F979A/N980A mutations in the bromodomain of TRIM24 inhibited TRIM24 association with H3K23ac, and also impaired the binding of TRIM24, H3K23ac

and STAT3 with ID1 promoter. Moreover, the association of TRIM24 with H3K23ac is important for EGFR-stimulated cell proliferation, cell migration, colony formation, tumor growth, GSC self-renewal, p-STAT3, ID1 expression and the binding of TRIM24, H3K23ac and STAT3 with ID1 promoter. This investigation identifies a previously unrecognized mechanism, in which H3K23ac/TRIM24 functions as a mediator of EGFR/EGFRvIII activation of STAT3 signaling, thereby promoting tumorigenesis in human cancers. Additionally, our results and the aforementioned studies also provide excellent evidence demonstrating the context-dependent functions of TRIM24 in modulating different cancers.

TRIM24 was reported as an oncogenic transcription factor in ER-driven breast cancer and AR-driven prostate cancer[8,19]. Here we demonstrate the oncogenic role of TRIM24 in GBM. Moreover, we show that TRIM24 functions as an oncogenic transcriptional co-activator of STAT3 and mediates EGFR/EGFRvIII-driven tumorigenesis. Aberrant EGFR signaling in GBM mediates STAT3 transcriptional activation to promote tumorigenesis, progression and invasion of glioblastoma[32,38]. Although a large body of knowledge has established the mechanisms by which EGFR activates STAT3 through JAK or Src[48], in this study, we show a novel mechanism by which activated EGFR enhances H3K23 acetylation and TRIM24 expression, promoting the association of TRIM24 with H3K23ac and STAT3, and facilitating STAT3 interaction with chromatin, leading to downstream signaling activation to drive glioma tumorigenesis. Our data demonstrate that the BBOX1 domain of TRIM24, specifically residues R193 and K195, is required for STAT3 binding and critical in STAT3 oncogenic signaling. R193A/K195A mutations in the TRIM24 BBOX1 domain inhibited STAT3 interactions and significantly reduced binding between TRIM24, H3K23ac and STAT3 within the ID1 promoter. R193A/K195A mutations also impaired EGFR-stimulated cell proliferation, cell migration, colony formation, tumor growth, GSC self-renewal, p-STAT3 and ID1 expression. These results suggest that EGFR/EGFRvIII-driven STAT3 activation depends on TRIM24-linked histone modification.

EGFR signaling activates the NF-κB transcription factor[49], an inflammatory signaling pathway, is also important in EGFR-driven glioma tumorigenesis[50–52]. Here, we show that although TRIM24 does not directly bind to the NF-κB p65 subunit, TRIM24 significantly altered the expression of NF-κB target genes in EGFRvIII-driven GBM cells, suggesting that TRIM24 regulates NF-κB in GBM cells in a manner distinct from STAT3. Our results further support that TRIM24 is required for EGFR/EGFRvIII-driven gliomagenesis.

In conclusion, our findings identify TRIM24 as an oncogenic transcriptional co-activator in EGFR-driven GBM and also demonstrate a previously unknown signal relay by which H3K23ac/TRIM24 mediates EGFR stimulation of STAT3 activation, thereby enhancing the oncogenic activity of the EGFR/STAT3 pathway in human cancers. The newly established roles of TRIM24 and H3K23ac in tumorigenesis provide a strong rationale for targeting them to treat glioma patients with high EGFR and STAT3 signaling activity.

## Methods

**Cell lines.** U87, LN229 and HEK293T cells were from ATCC (Manassas, VA, USA). Patient-derived glioma stem cell (GSC) lines, GSC1123 and GSC83 were from Dr. Ichiro Nakano[42]. GSC cells were maintained in DMEM/F12 supplemented with B27 (1 : 50), heparin (5 mg/ml), basic FGF (20 ng/ml), and EGF (20 ng/ml), and glioma cells were maintained in Dulbecco's Modified Eagle's Medium (DMEM) supplemented with 10% fetal bovine serum as we previously described[42,44]. All cell lines were cultured at 37 °C and 5% CO$_2$. U87 and LN229 cell lines were also recently authenticated using a STR DNA fingerprinting at Shanghai Biowing Applied Biotechnology Co., Ltd (Shanghai, China), and

mycoplasma infection was detected using LookOut Mycoplasma PCR Detection kit (Sigma-Aldrich). U87/EGFRvIII and LN229/EGFRvIII cell lines that overexpress exogenous EGFRvIII were established by transducing EGFRvIII into U87 and LN229 cells and characterized using WB as previously described[27,53].

**Antibodies and reagents.** The following antibodies were used in this study: anti-β-actin (I–19, 1 : 500), anti-Histone H3 (FL-136, 1 : 500) and anti-STAT3 (H-190, 1 : 1000) antibodies (Santa Cruz Biotechnology); a monoclonal anti-Flag M2 antibody (Sigma-Aldrich, 1 : 1000); anti-EGFR antibody (D38B1, 1 : 1000), anti-phospho-EGFR (Y1173)(53A5, 1 : 1000), anti-Tri-Methyl-Histone H3 (Lys4) (#9727, 1 : 1000), anti-Tri-Methyl-Histone H3 (Lys27)(C36B11, #9733, 1 : 1000), anti-STAT3 (124H6, #9139, 1 : 1000), anti-Acetyl-Histone H3 (Lys27) (#4353, 1 : 1000), anti-NF-κB p65 (#3034, 1 : 1000), and anti-phospho-STAT3 (Tyr705) (D3A7, 1 : 1000) antibodies (Cell Signaling Technology); an anti-ID1 antibody (ab168256, Abcam, 1 : 500); an anti-acetyl-Histone H3 (Lys23) antibody (#07-355, 1 : 1000, Millipore-Upstate); an anti-TRIM24 antibody (#14208-1-AP, 1:500, Proteintech Group). The secondary antibodies were from Vector Laboratories or Jackson ImmunoResearch Laboratories. Peroxidase blocking reagent was from DAKO. AquaBlock was from East Coast Biologics, Inc. Erlotinib was from LC Laboratories. Cell culture media and other reagents were from Invitrogen, Sigma-Aldrich, VWR, or ThermoFisher Scientific.

**Plasmids.** Flag-TRIM24 was a gift from Michelle Barton (Addgene plasmid # 28138)[18], and pLEGFP-WT-STAT3 was a gift from George Stark (Addgene plasmid # 71450)[54]. Then, TRIM24 was subcloned and inserted into a lentivirus pLVX-Puro vector (Clontech) or a pcDNA3 vector (Invitrogen). TRIM24$^{F979A/N980A}$ and TRIM24$^{R193A/K195A}$ point mutations were generated using a site-directed mutagenesis kit (Invitrogen) following the manufacturer's protocol. TRIM24 shRNAs were purchased from Genechem (Shanghai, China). ID1 cDNA was amplified by PCR from normal human brain tissues, and then subcloned into a lentivirus LeGO-iG vector. LeGO-iG was a gift from Boris Fehse (Addgene plasmid# 27358).

**Immunoprecipitation (IP) and Western blotting (WB) assays.** WB and IP analyses were performed as we previously described[26]. Briefly, various cells were lysed in an IP buffer (20 mM Tris-HCl, pH 7.5, 150 mM NaCl, 1 mM EDTA, 2 mM Na3VO4, 5 mM NaF, 1%Triton X-100 and protease inhibitor cocktail) at 4 °C for 30 min. The lysates were centrifuged for 20 min at 12,000 × g. Protein concentrations were determined, and then equal amounts of cell lysates were immunoprecipitated with specific antibodies and protein G-agarose beads (Invitrogen). Immunoprecipitates were washed five times with IP buffer, resolved in a 2× SDS lysis buffer and analyzed in a SDS-PAGE gel. Uncropped scans of WB presented in the main text of this study (Figs. 1–8) are included in as the Supplementary Fig. 9 in the Supplementary Information in this paper.

**Cell proliferation and migration assays.** Cell proliferation analysis was performed using a WST-1 assay kit (Roche) as previously described[27]. Cells were seeded in medium, split, and detected with a WST-1 assay kit. Cell migration analysis was performed using a Boyden chamber as previously described[55]. Various cells were serum-starved for 24 h, washed with PBS and resuspended in DMEM plus 0.1% FBS. Then, cells were placed into the top compartment of a Boyden chamber and the bottom chamber was added with 10% FBS/DMEM. The cells were allowed to migrate through an 8-μm pore size membrane precoated with fibronectin (10 μg/ml) for 10 or 16 h at 37 °C. Afterwards, the membrane was fixed, stained and analyzed.

**Colony formation assay.** Soft agar colony formation assay was performed as we previously described[56]. Briefly, cells were seeded in a 0.4% Noble Agar top layer with a bottom layer of 0.8% Noble Agar in each of the triplicate wells of a 24-well plate. Cell culture media was added 3 days after plating and changed every 3 days thereafter. Colonies were scored after 2–3 weeks using Olympus SZX12 stereomicroscope, and data were analyzed using GraphPad Software.

**shRNA knockdown and transfection.** These assays were performed as previously described[57]. Lentiviruses were produced by co-transfecting various cDNA and packaging plasmids into HEK293T cells using Lipofectamine 2000 reagent according to manufacturer's instruction (#52758, Invitrogen). Forty-eight hours after transfection, the supernatants containing viruses were filtered and added into the culture media supplemented with 8 μg/ml polybrene. Transduced human GBM cells or glioma stem cells were collected at least 48 h post-infection, and protein expression of exogenous genes were validated by WB.

**Promoter reporter and dual luciferase assays.** TRIM24 promoter was amplified using PCR with primers (5′-TAGCCCGGGCTCGAGGATCAAATTA-CATGGAATTCTTTCAAAAC-3′ and 5′-CGGAATGCCAAGCTTAGCGGA GACCGTTCCTCGCACC-3′), and then inserted into pGL3-Basic vector. For normalization of transfection efficiency, pRL-TK (Renilla luciferase) reporter plasmid was added to each transfection. The activites of firefly luciferase and

Renilla luciferase were quantified using a dual-specific luciferase assay kit according to manufacturer's instruction (#E1910, Promega).

**RNA-Seq and Differentially expressed gene analysis**. Total RNA was extracted and purified using the Qiagen RNeasy Mini kit (Valencia, CA, USA) according to the manufacturer's instructions. The quality of RNA was assessed by a bioanalyzer before sequencing. Libraries for poly(A)$^+$ RNA were prepared according to the Illumina protocol. Libraries were sequenced on Illumina HiSeqX Ten platforms. The criteria of Differentially Expressed Genes detection in this study are false discovery rate (FDR) <0.05 and a fold change >1.5. Expression patterns were clustered with Cluster 3.0 and viewed using Java Tree View 3.0.

**Quantitative Real-Time PCR analysis**. Quantitative Real-Time PCR was performed in triplicate using the QuantiTect SYBR Green PCR Kit (Qiagen, Valencia, CA, USA) on a Rotorgene 6000 series PCR machine (Corbett Research, Valencia, CA, USA). All mRNA quantification data were normalized to *ACTB*, which was used as an internal control. Primers for *ID1*: 5′-GTAAACGTGCTGCTCTAC-GACATGA-3′ and 5′-AGCTCCAACTGA AGGTCCCTGA-3′. Primers for *ID3*: 5′-GAGAGGCACTCAGCTTAGCC-3′ and 5′-TCCTTTTGTCGTTGGA-GATGAC-3′. Primers for *FGFR3*: 5′-TGCGTCGTGGAGAACAAGTTT-3′ and 5′-GCACGGTAACGTAGGGTGTG-3′. Primers for *TGFA*: 5′-AGGTCCGAAAA-CACTGTGAGT-3′ and 5′-AGCAAGCGGTTCTTCCCTTC-3′. Primers for *AURKA*: 5′-GAGGTCCAAAACGTGTTCTCG-3′ and 5′-ACAGGATGAGGTA-CACTGGTTG-3′. Primers for *DDX39A*: 5′-GCAGATTGAGCCTGTCAACG-3′ and 5′-GACCACCGAAGAACACAGAC-3′. Primers for *ACTB*: 5′-CATG-TACGTTGCTATCCAGGC-3′ and 5′-CTCCTTAATGTCACGCACGAT-3′. Primers for *TRIM24*: 5′-TGTGAAGGACACTACTGAGGTT-3′ and 5′-GCTCTGATACACGTCTTGCAG-3′.

**ChIP-qPCR**. ChIP was performed using the Chromatin Immunoprecipitation Kit (Millipore-Upstate) according to the manufacturer's instructions. Immunoprecipitated DNA was purified after phenol extraction and was used for qRT-PCR. Primers of *ID1* (promoter): 5′-GAGGGAGACCCTGCTCGA-3′ and 5′-GCAGTGGAGTGAGGCTGCA-3′.

**Limiting dilution neurosphere-forming assay**. Limiting dilution assay was performed as we previously described[44]. In brief, dissociated cells from glioma spheres were seeded in 96-well plates containing GSC culture medium (20–200 cells per well). After 7 days, each well was examined for formation of tumor spheres. Stem cell frequency was calculated using extreme limiting dilution analysis (http://bioinf.wehi.edu.au/software/elda/).

**Tumorigenesis studies**. All animal experiments were performed in accordance to a protocol approved by Shanghai Jiao Tong University Institutional Animal Care and Use Committee (IACUC). Athymic (Ncr nu/nu) female mice at an age of 6–8 weeks (SLAC, Shanghai, China) were used. Mice were randomly divided into 5–6 per group. Four thousand patient-derived GSCs or $5 \times 10^5$ human GBM cells were stereotactically implanted into the brain of the animals as we previously described[26,44]. Mice were euthanized when neuropathological symptoms developed. Two separate individuals who were blinded to measure tumor volumes as $(W^2 \times L) / 2$, $W < L$[58].

**IHC of human GBM specimens**. In accordance to a protocol approved by Shanghai Jiao Tong University Institutional Clinical Care and Use Committee, 125 primary human GBM specimens were collected from 2001 to 2015 at Ren Ji Hospital, School of Medicine, Shanghai Jiao Tong University, Shanghai, China. The informed consent was obtained from all patients. These clinical cancer specimens were examined and diagnosed by pathologists at Ren Ji Hospital, School of Medicine, Shanghai Jiao Tong University. The tissue sections from paraffin-embedded de-identified human GBM specimens were stained with antibodies against p-EGFR (p-EGFR$^{Y1173}$)(1:50), TRIM24 (1:200), acetyl-Histone H3 (Lys23)(1:200) and p-STAT3 (p-STAT3$^{Y705}$)(1:50). Non-specific IgGs were used as negative controls. IHC staining was quantified as previously described[57] and modified as follows: 7, strong staining in ~50% of tumor cells; 6, weak staining in ~ 50% of tumor cells; 5, strong staining in ~25% of tumor cells; 4, weak staining in ~ 25% of tumor cells; 3, strong staining in ~5 to 25% of tumor cells; 2, weak staining in ~5–25% of tumor cells; 1, low or no staining in < 1% of tumor cells; 0, no detectable staining in all tumor cells (0%). Tumors with 0 to 2 staining scores were considered as low expressing and those with 3 to 7 staining scores were considered high expressing. Two separate individuals who were blinded to the slides examined and scored each sample. Analyses of Spearman's rank correlation and Kaplan-Meier survival were performed as previously described[27].

**Statistics**. GraphPad Prism version 5.0 for Windows (GraphPad Software, Inc., San Diego, CA, USA) was used to perform one-way analysis of variance (ANOVA) with Newman-Keuls post-hoc test or an unpaired, two-tailed Student's *t*-test. Kaplan-Meier survival analysis was carried out using log-rank tests. A Spearman's rank correlation analysis was used to investigate the correlation of protein expression levels in human clinical GBM specimens. A *P*-value of less than 0.05 was considered significant.

**Study approval**. All the work related to human tissues was performed at the Shanghai Jiao University under institutional review board (IRB)-approved protocols, according to NIH guidelines. All experiments using animals were performed at the Shanghai Jiao University under the Institutional Animal Care and Use Committee–approved protocols, according to NIH guidelines.

**Data availability**. All relevant data are available from the authors. RNA-Seq data reported in this study have been deposited with the Gene Expression Omnibus under the accession GEO ID: GSE95386. All the other data supporting the finding of this study are available within the article and its Supplementary Information files or from the corresponding author on reasonable request.

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

## Acknowledgements

We thank Ichiro Nakano for providing patient-derived glioma stem cells. This work was supported in part by National Natural Science Foundation of China (No. 81372704, 81572467 to H.F.; No.81470315, 81772663 to Y.L.); the Program for Professor of Special Appointment (Eastern Scholar) at Shanghai Institutions of Higher Learning (No. 2014024), Shanghai Municipal Education Commission—Gaofeng Clinical Medicine Grant Support (No. 20161310), New Hundred Talent Program (Outstanding Academic Leader) at Shanghai Municipal Health Bureau (2017BR021), Technology Transfer Project of Science & Technology Dept. at Shanghai Jiao Tong University School of Medicine (ZT201701) to H. Feng; Shanghai Jiao Tong University School of Medicine Hospital Fund (No. 14XJ10069), Collaborative Innovation Center for Translation Medicine at Shanghai Jiao Tong University School of Medicine (TM201502) to Y.L.; Shanghai Natural Science Foundation (16ZR1420200) and Shanghai Jiao Tong University Medical Engineering Cross Fund (YG2015QN35) to W.Z.; US NIH grants (CA158911, NS093843, NS95634 and CA209345), a Zell Scholar Award from the Zell Family Foundation and funds from Northwestern Brain Tumor Institute at Northwestern University to S.-Y.C.; a Brain Cancer Research Award from James S. McDonnell Foundation to B.H.; an NIH/NCI training grant T32 CA070085 to A.A.

## Author contributions

D.L., Y.L., and H.F. designed the experiments. D.L., Y.L., W.Z., L.S., and J.T. performed experiments. D.L., Y.L., and H.F. interpreted the data. D.L., Y.L., W.-Q.G., A.A.A., B.H., S.C., and H.F. wrote and reviewed the manuscript. H.F. supervised the project.

## Additional information

**Competing interests:** The authors declare no competing financial interests.

