## [Peer Review File · Nature Communications]

Reviewers' comments:

Reviewer #1 (Remarks to the Author):

In this manuscript, Lv and colleagues demonstrated that TRIM24 functions as oncogenic transcriptional co-activator for STAT3 activation in glioblastoma. The authors provide the evidence for that EGFR signaling promotes H3K23 acetylation and association with TRIM24; and TRIM24 functions as a co-activator to recruit STAT3, leading to stabilized STAT3-chromatin interactions and subsequent activation of STAT3 downstream signaling, thereby enhancing EGFR-driven tumorigenesis. This work is interesting and significant for GBM cancer research. The paper is clearly presented and well written, and the experiments are performed in a logical way to address their major conclusions. The biochemical and functional data are very strong throughout and serve to support the claims in most cases except of a few minor points that need to be improved:

1. The method for quantification of IHC staining of human GBM specimens only take percentage of positive cells into consideration, but not staining intensity. More clarification is needed.
2. The subtypes of the GSCs GSC1123 and GSC83 GSCs are unknown? More detail information would be helpful.
3. Figure 9, several arrows were included to indicate positive staining. However, it seems that much more cells are positive. Thus, the arrows should be removed.
4. In the Discussion section, whether other downstream EGFR transcription factors, such as c-Jun, are affected by TRIM24 should be discussed.

Reviewer #2 (Remarks to the Author):

Here, Lv et al., perform a series of genetic and biochemical studies in GBM cell lines, GSC models and in vivo models, identifying a previously unknown role for TRIM24 in GBM pathogenesis. They propose that aberrant EGFR promotes GBM pathogenesis by regulating TRIM24 level, which acts as a transcriptional co-activator recruiting STAT-3 to chromatin to drive a gene expression program that promotes tumorigenesis. The biochemical studies are strong; the correlative data to clinical samples adds strength and relevance, and the mechanistic studies are also by in large convincing. The insights are valuable and the data largely support the conclusions.

However, a number of issues are raised which if addressed, would markedly strengthen the paper.

1) Specificity of TRIM24 for STAT-3: the paper hinges on showing that STAT-3 regulated genes are under control of TRIM24 through alteration of chromatin. This may very well be true. However, to lend confidence, a few more experiments would be helpful. The authors focus on 6 genes identified by shRNA knockdown of TRIM24 in one of the cell lines, and then focus specifically on ID1, a known STAT-3 target. However, the altered set of genes is potentially much broader. EGFR alters many transcription factors and it is possible that the genes dysregulated by TRIM24 knockdown include genes controlled by other TFs? Does the TRIM24 knockdown signature closely resemble a STAT-3 knockdown signature, or does it suggest that STAT-3 is one of the components? Either answer is acceptable and interesting, but the specificity for STAT-3 needs to be determined.

2) How does EGFRvIII/EGFR regulate TRIM24? I assume it is transcriptional - is this correct? If so, which signaling pathway or pathways and, is it possibly itself regulated by STAT-3?

Overall, the MS is interesting and potentially important. Addressing these two issues would markedly strengthen the paper.

One minor issue - Figure 7 f-j is very hard to read.

Point-to-point response to the reviewers' comments

Reviewer #1:

1. The method for quantification of IHC staining of human GBM specimens only take percentage of positive cells into consideration, but not staining intensity. More clarification is needed.

Response: We thank the reviewer for this excellent suggestion. We have modified the method for quantification of IHC staining of human GBM specimens as follows in page 29: 7, strong staining in ~50% of tumor cells; 6, weak staining in ~50% of tumor cells; 5, strong staining in ~25% of tumor cells; 4, weak staining in ~25% of tumor cells; 3, strong staining in ~5 to 25% of tumor cells; 2, weak staining in ~5 to 25% of tumor cells; 1, low or no staining in <1% of tumor cells; 0, no detectable staining in all tumor cells (0%). Tumors with 0 or 1 staining scores were considered as low expressing and those with 2 to 7 staining scores were considered high expressing. Kaplan-Meier analyses were performed again in Fig.9 and Supplementary Fig. 8.

2. The subtypes of the GSCs, GSC1123 and GSC83 GSCs are unknown? More detail information would be helpful.

Response: GSC1123 and GSC83 are glioma stem cells that were previously characterized as a mesenchymal subtype. This detail along with the citation was added in page 16.

3. Figure 9, several arrows were included to indicate positive staining. However, it seems that much more cells are positive. Thus, the arrows should be removed.

Response: We thank the reviewer for this excellent suggestion. The arrows have been removed in Fig. 9.

4. In the Discussion section, whether other downstream EGFR transcription factors, such as c-Jun, are affected by TRIM24 should be discussed.

Response: We thank the reviewer for this excellent suggestion. We have performed GSEA analysis and found that TRIM24 regulates STAT3 signaling pathway and NF- κ B signaling pathway in Fig.6a and Supplementary Fig.5a. However, IP and WB assays showed that TRIM24 did not bind with the NF- κ B subunit p65 (Supplementary Fig.5a). Our data suggest that TRIM24 functions as a transcriptional co-activator of STAT3 but not NF- κ B. We have discussed this in page 22.

Reviewer #2 (Remarks to the Author):

1) Specificity of TRIM24 for STAT-3: the paper hinges on showing that STAT-3 regulated genes are under control of TRIM24 through alteration of chromatin. This may very well be true. However, to lend confidence, a few more experiments would be helpful. The authors focus on 6 genes identified by shRNA knockdown of TRIM24 in one of the cell lines, and then focus specifically on ID1, a known STAT-3 target. However, the altered set of genes is potentially much broader. EGFR alters many transcription factors and it is possible that the genes dysregulated by TRIM24 knockdown include genes controlled by other TFs? Does the TRIM24 knockdown signature closely resemble a STAT-3 knockdown signature, or does it suggest that

STAT-3 is one of the components? Either answer is acceptable and interesting, but the specificity for STAT-3 needs to be determined.

Response: We thank the reviewer for this excellent suggestion. As Reviewer #1 requested, we have performed GSEA analysis and found that TRIM24 regulates both STAT3 and NF- κ B signaling pathways in Fig.6a and Supplementary Fig.5a. However, IP and WB assays showed that TRIM24 did not bind with NF- κ B (Supplementary Fig.5a). We reference this data on page 12. With respect to the STAT3 target gene ID1, we demonstrate EGFRvIII/TRIM24/STAT3 signaling pathway regulated ID1 activity on multiple levels using gene expression microarrays (Fig. 5a), ChIP-qPCR (Fig. 6c, 6d, 7j), qRT-PCR (Fig. 7f, Supplementary Fig. 4), and Western blotting (Fig. 7e, 8a, 8b, Supplementary Fig. 7b, 7c), and validate the importance of ID1 in our glioma lines (Supplementary Fig. 7). Our data suggest that TRIM24 functions as a transcriptional co-activator of STAT3 but not NF- κ B.

2) How does EGFRvIII/EGFR regulate TRIM24? I assume it is transcriptional - is this correct? If so, which signaling pathway or pathways and, is it possibly itself regulated by STAT-3?

Response: We thank the reviewer for this excellent suggestion. We have performed new assays and found that TRIM24 is transcriptionally regulated by STAT3 in EGFR/EGFRvIII-driven GBM cells in Supplementary Fig.6a-6e. We also have described this in page 12.

3) One minor issue - Figure 7 f-j is very hard to read.

Response: We apologize for this oversight. We have revised the figure and modified the colors for better clarity.

REVIEWERS' COMMENTS:

Reviewer #1 (Remarks to the Author):

The revised manuscript has addressed the concerns from this reviewer

Reviewer #2 (Remarks to the Author):

The authors have done a commendable job of addressing the critiques. The paper is strengthened by these new data and will be a valuable contribution to the literature.